# Uptake of cervical cancer screening and its determinants in Africa: Umbrella review

Berihun Agegn Mengistie[1]*, Mihret Melese[2], Ashebir Mamay Gebiru[3], Mihret Getnet[2], Amare Belete Getahun[4], Worku Chekol Tassew[5], Mikias Mered Tilahun[6], Yosef Belay Bizuneh[4], Habtu Kifle Negash[7], Nebebe Demis Baykemagn[8], Desale B. Asmamaw[9], Amlaku Nigusie Yirsaw[10], Alemken Eyayu Abuhay[11], Desalegn Anmut Bitew[9]

1 Department of General Midwifery, School of Midwifery, College of Medicine and Health Sciences, University of Gondar, Gondar, Ethiopia, 2 Department of Human Physiology, School of Medicine, College of Medicine and Health Sciences, University of Gondar, Gondar, Ethiopia, 3 Department of Health Informatics, Teda Health Science College, Gondar, Ethiopia, 4 Department of Anesthesia, College of Medicine and Health Sciences, University of Gondar, Gondar, Ethiopia, 5 Department of Medical Nursing, Teda Health Science College, Gondar, Ethiopia, 6 Department of Optometry, School of Medicine, College of Medicine and Health Sciences, University of Gondar Comprehensive Specialized Hospital, Gondar, Ethiopia, 7 Department of Human Anatomy, School of Medicine, College of Medicine and Health Sciences, University of Gondar, Gondar, Ethiopia, 8 Department of Health Informatics, Institute of Public Health, College of Medicine and Health Sciences, University of Gondar, Gondar, Ethiopia, 9 Department of Reproductive Health, Institute of Public Health, College of Medicine and Health Sciences, University of Gondar, Gondar, Ethiopia, 10 Department of Health Promotion and Health Behavior, Institute of Public Health, College of Medicine and Health Sciences, University of Gondar, Gondar, Ethiopia, 11 University of Gondar Comprehensive Specialized Hospital, Gondar, Ethiopia

* berihunagegn21@gmail.com

## Abstract

### Background

Cervical cancer is the fourth most prevalent type of cancer in women globally. Early detection and treatment of precancerous cervical lesions and human papillomavirus (HPV) infection are strongly advised to decrease the incidence of cervical cancer and death. Cervical cancer is a major public health concern in low- and middle-income nations, where screening and treatment options are constrained. Thus, the main objective of this umbrella review was to determine the pooled uptake of cervical cancer screening and its determinants in Africa.

### Methods

This study followed the Preferred Reporting Items for Systematic Reviews and Meta-Analyses (PRISMA) guidelines. The protocol for this umbrella review was registered on the International Prospective Register of Systematic Reviews (PROSPERO) with reference number CRD42024518297. We conduct a systematic and comprehensive search by using Google Scholar, PubMed, Scopus, Hinari, and Science Direct, from January 1, 2014, to September 20, 2024. The data were extracted

**Data availability statement:** All relevant data are within the paper and its Supporting Information files.

**Funding:** The author(s) received no specific funding for this work.

**Competing interests:** The authors have declared that no competing interests exist.

**Abbreviations:** AOR, Adjusted odds ratio; AMSTAR, A Measurement tool to Assess systematic reviews; CCS, Cervical cancer screening; CI, Confidence interval; HPV, Human papillomavirus; LMICs, Low-and middle-income countries; SSA, sub-Saharan Africa; STIS, Sexually transmitted infections; SRM, Systematic review and meta-analysis; WHO, World Health Organization.

using Microsoft Excel spreadsheet. The methodological quality of the included studies was examined using A Measurement Tool to Assess Systematic Reviews 2 (AMSTAR 2). The statistical analysis was carried out using STATA version 17, which includes descriptive analysis, forest plots for prevalence, funnel plot, and an Egger test to examine publication bias. A random-effects model was used to determine the pooled effect estimate. Publication bias was checked by using the funnel plot and Egger's tests.

## Results

This umbrella review included 11 systematic reviews and meta-analysis studies across Africa with a total of 143,327 study participants. The overall prevalence of cervical cancer screening practice in Africa was 20.94% (95% CI: 15.84%–26.04%). Women's level of knowledge (AOR: 3.22, 95% CI: 1.64–6.33), positive attitude toward CCS (AOR: 2.48, 95% CI: 2.18–2.81), perceived vulnerability to cervical cancer (AOR = 3.57, 95% CI: 2.75, 4.63), and history of STIs (AOR = 4.89, 95% CI: 3.14, 7.62) were significantly associated with cervical cancer screening practice. In conclusion, the combined estimate of cervical cancer screening use in Africa remains much lower (20.94%) than the World Health Organization (WHO) recommendations target (70%). It indicates that there is a large gap that requires being addressed in collaboration to reduce the burden of cervical cancer and its morbidity and mortality across the continent. Therefore, healthcare professionals, policymakers, and other stakeholders shall implement effective strategies such as empowering women, improving the knowledge and attitude towards cervical cancer screening, advocacy, and expanding screening programs to all eligible women to increase utilization of cervical cancer screening.

## Introduction

Cervical cancer affects mainly women and girls worldwide [1]. It's the fourth most prevalent cancer in both incidence (6.8%) and mortality (8.1%) in women worldwide in 2022 [1]. Even though it's a global public health problem, the incidence is higher in low- and middle-income countries [1,2]. In 2022, cervical carcinoma was estimated to cause 661,021 cases of new cancer and 348,189 mortalities globally [1]. It's a rare cause of death in developed nations, while about 94% of the deaths from cervical cancer occurred in low- and middle-income nations [1,2]. The incidence and fatality rates for cervical cancer remain highest in Sub-Saharan Africa (SSA), Central America, and Southeast Asia [1,2]. Geographic variations in the cervical cancer burden are linked to disparities in access to HPV immunization, cervical cancer screening, and treatment services [1].

Human papillomavirus (HPV) infection causes nearly 99.7% of precancerous and malignant cervical lesions [3,4]. HPV subtypes 16 and 18, which account for approximately 70% of cervical cancer around the globe, are the most carcinogenic types [3].

Cervical cancer is caused by a persistent sexually transmitted infection of oncogenic HPV variants [4,5]. Additionally, there are several common risk factors that raise the likelihood of developing cervical cancer, including having multiple sexual partners, initiation of early sexual intercourse, having a compromised immunity, using oral contraception for an extended period of time (more than 5 years), having a large number of children (three or more), poor genital sanitation, tobacco use, and having sexually transmitted infections [6,7].

Worldwide, the uptake of cervical cancer screening (CCS) varies greatly [8]. High-income countries have higher rates, while low-income ones have lower rates. In 2019, the global adherence of CCS in women aged 20–69 years was 33.66% [8]. This rate was twice as higher in developed nations, at 75.66%, as compared to low-and middle-income countries (LMICs), which reported only 24.91% compliance. Similarly, CCS uptake is extremely low in Africa (5.28%) [8]. Consequently, the region of Sub-Saharan Africa (SSA) ranks as the highest regarding poor cervical cancer outcomes, such as higher mortality and lower survival rate, suggesting late diagnosis of cases, limited access for screening and treatment services, and the presence of additional risk factors, including a high burden of HIV and poor socioeconomic factors and extreme poverty [9,10]. In addition, A wide range of barriers, such as lack of knowledge and awareness of cervical cancer, cultural or traditional and religious factors, and health system barriers to screening, were identified across most LMICs [11].

The WHO has launched a Global Strategy to Accelerate Cervical Cancer Elimination that defines three important strategies: HPV vaccination, screening, and treatment [9]. In 2020, the WHO introduced the 90-70-90 a global strategy to eliminate cervical cancer. This plan targets 90% HPV vaccination of girls by the age of 15, 70% of women to be screened for cervical disease with a high-performance test at least twice by the age of 45. Finally, 90% of women diagnosed with precancerous cervical lesions treated and 90% of invasive cervical cancer managed by 2030 [9]. The primary objective of secondary prevention is to lower the incidence and mortality of cervical cancer by detecting and treating women who have precancerous lesions [9].

Cervical cancer grows slowly from HPV infection or precancerous lesions to advanced stages of malignancy. It can be prevented using screening, thereby allowing for early identification and an opportunity for therapy [12]. Screening for cervical cancer is one of the most effective cancer preventive techniques available today. If abnormal changes to cervical epithelial cells are identified and treated immediately, the likelihood of developing cervical cancer is lowered [13]. The American Cancer Society (ACS), the WHO, and the United States Preventive Services Task Force (USPSTF) suggest that all eligible women receive a screening for cervical cancer at least once every three years [14]. There are three types of cervical cancer screening methods [15]. HPV DNA testing is recommended as the primary screening technique for women older than 30 years. The second method is Visual Inspection with Acetic Acid (VIA) or Visual Inspection with Lugol's (VILI), which is used when HPV testing is not yet available or when there is a risk of loss to follow-up. The third approach is the Pap smear, which should be performed in the following situations: 1) for women who are not qualified for VIA or VILI due to the non-visibility of the squamo-columnar junction and absence of HPV screening, 2) as the main test for women younger than 30 years, and 3) as a co-test with HPV for HIV-infected women [15,16]. In resource-limited areas, visual inspection of the cervix with acetic acid (VIA) followed by therapy (screen and treat) is an alternative approach to secondary prevention [9,16].

Although cervical cancer screening is proven to reduce cervical cancer incidence, many factors influence screening uptake [17]. The rate of screening uptake has been shown to vary by knowledge about cervical cancer and screening services, in addition to other factors, such as individual perception, beliefs, attitudes, culture, and partner attitude [18]. Numerous individuals or coordinated initiatives targeting both health practitioners and the community may encourage women to undergo CCS. Outreach visits, mobilization of communities, health education, and counseling services are all possible components of an efficient cervical cancer prevention and control program that ensures high screening uptake [19,20].

Despite cervical cancer screening being considered one of the most efficient approaches for reducing the incidence and mortality rate of cervical cancer, it's less practiced in LMIC countries, including Africa [9]. So far, there are multiple

systematic reviews and meta-analyses that have found an inconsistent prevalence of CCS uptake with significant hetero-geneity across Africa, ranging from 5.47% [21] to 43% [22]. In addition, this umbrella review attempts to aggregate the overall estimate of CCS uptake and its determinants among different study populations. For instance: eligible-age group women, reproductive-age women, and women living with HIV. Therefore, the aim of this umbrella review was to summa-rize the heterogeneous findings of various systematic reviews and meta-analysis studies to generate conclusive findings. This comprehensive evidence will help HCPs, health managers, and policymakers in implementing evidence-based inter-ventions to improve uptake of CCS, and it's considered a cost-effective secondary preventive intervention for cervical can-cer. Ultimately, it enables the detection and treatment of precancerous cervical lesions, improves survival rates, reduces morbidity and mortality from cervical cancer.

### Research questions

1. What is the pooled estimate of cervical cancer screening uptake among women in Africa?

2. What are the determinant factors of cervical cancer screening uptake among women in Africa?

## Methods and materials

### Study protocol and search strategy

This umbrella review followed the Preferred Reporting Items for Systematic Reviews and Meta-Analysis Protocols (PRISMA-P) guideline [23] (S1 File). The study protocol was developed and registered on PROSPERO (reference number: CRD42024582302).

We conducted a comprehensive search on Google Scholar, PubMed, Hinari, Scopus, and ScienceDirect. on system-atic review and meta-analysis studies of CCS uptake and its determinant factors undertaken in Africa. The search tech-nique was based on the condition, context, and population (CoCo Pop) framework [24]. Publications were retrieved from prior studies meeting the eligibility criteria. A search strategy was developed for databases by combining keywords using Boolean operators ("AND" and "OR"). All systematic reviews and meta-analyses published between January 1, 2014, and September 20, 2024, were included. Finally, we used the following combination of searching terms: ("systematic review AND meta-analysis" OR "systematic reviews") AND "uptake" OR "acceptance" OR "utilization" OR "cervical cancer screen-ing" OR "early cervical cancer detection" AND "Africa" OR "Sub-Saharan Africa." In addition, snowballing techniques were used to retrieve further studies from the citation list of papers identified in the available database (S2 File).

### Eligibility criteria

We included all systematic reviews and meta-analyses studies that reported the prevalence and/or associated factors of cervical cancer screening in Africa. In addition, studies published in English after January 1, 2014, were included in this umbrella review. Whereas articles were excluded for the following reasons: the article did not report the outcome of interest, narrative reviews, primary studies, qualitative reviews, expert opinions, case reports, editorials, correspondence, abstracts, and methodological studies.

### Data extraction and management

Two authors (BAM and MM) carried out data extraction from the included SRM studies using a standardized data abstrac-tion form developed in an Excel spreadsheet. Based on the inclusion and exclusion criteria, all searched studies were transferred to Endnote X8, reference management software. Articles were screened and selected first based on their title and abstract, and then the full text was reviewed. In cases of dispute, discussions with additional reviewers were held to determine the final article selection to include in this umbrella review. Following the comprehensive searching, possibly

eligible publications were imported into Endnote. Duplicate studies were deleted in cases where two or more papers shared similar features. Structured data extraction in a Microsoft Excel spreadsheet was designed and implemented. For each SRM study, the following data were extracted: identification data (first author's last name and publication year), prevalence of CCS, factors associated with CCS uptake, adjusted odds ratio with 95% confidence intervals, number of primary studies included within each SRM study, study area, total sample size, publication bias assessment methods, risk bias assessment method, and scores (Table 2 and S3 File).

### Measurement of outcome variable

The primary objective of this study was to determine the combined prevalence of cervical cancer screening uptake in Africa. The prevalence was calculated by dividing the number of women who screened for cervical cancer by the total number of women in the study and then multiplying by 100. The second objective of this umbrella review was to identify predictors of CCS uptake in Africa, which were evaluated using adjusted odds ratios from previous SRM studies.

Cervical cancer screening is the practice of checking women for precancerous or malignant cells on the cervix using diagnostic procedures including the Pap smear, HPV test, or visual inspection with acetic acid (VIA). These procedures involve taking cervical cells for microscopic analysis or diagnosing the presence of HPV, a virus associated with cervical cancer [9,25]. For the purposes of this analysis, any documented evidence of having screened and participated in mass or campaign examinations, clinic- or facility-based, or regular screenings a minimum of once in life is considered CC screening uptake [9,26].

### Quality assessment

The quality of the included studies was evaluated by two authors (BAM and MM) independently using A Measurement Tool to Assess Systematic Reviews (AMSTAR 2) [27]. The tool consists of 16 components, comprising 9 non-critical and 7 critical subdomains. Critical domains include whether the protocol was registered prior to the beginning of the review, the scope of the literature search, the explanation for excluding specific studies, the risk of bias from the studies included in the review, the suitability of meta-analysis methods, performing the risk of bias account of when interpreting the review's findings, and the evaluation of the existence and potential effects of publication bias [27]. The quality of included SRM studies in this umbrella review was classified as high, moderate, low, or critically low using the AMSTAR-2 tool. This umbrella review turned down articles with critically low-quality evidence [27] (Table 1).

### Data synthesis and statistical analysis

Data were extracted using a Microsoft Excel spreadsheet and then exported to STATA 17 statistical software, where all statistical data analysis were performed. The extracted data were presented as texts, tables, and forest plots. The standard error of prevalence for each SRM study was calculated using a binomial distribution. The pooled prevalence of the SRM studies was examined for heterogeneity using the Higgins I-squared ($I^2$) test. Heterogeneity among those included was characterized as low, moderate, or high based on I-square values of <25%, 50%−75%, and 75%, respectively [28].

A random-effects meta-analysis approach (Der Simonian and Laird's method) was employed to estimate the pooled prevalence of CCS use in Africa. Subgroup analysis was performed across the country, continent, and study population to identify potential sources of study heterogeneity. Additionally, we conducted a leave-one-out sensitivity analysis to assess the impact of individual SRM studies on the pooled estimate. The pooled estimates across the continent were then displayed using forest plots and tables, along with their respective 95% confidence intervals. Graphically, publication bias was examined using a forest plot (23). Furthermore, the statistical significance of publication bias was tested using both Egger's and Begg's tests, and a p-value less than 0.05 was employed to confirm the existence of publication bias [29]. In the end, the pooled adjusted odds ratio (AOR) with 95% confidence intervals was displayed using forest plots.

**Table 1. Methodological quality assessment of included systematic and meta-analysis studies using AMSTAR criteria.**

| Authors | Q1 | Q2 | Q3 | Q4 | Q5 | Q6 | Q7 | Q8 | Q9 | Q10 | Q11 | Q12 | Q13 | Q14 | Q15 | Q16 | Score |
|---|---|---|---|---|---|---|---|---|---|---|---|---|---|---|---|---|---|
| Yimer et al., 2021 | N | Y | Y | Y | N | Y | Y | Y | Y | Y | Y | Y | Y | Y | Y | Y | 15 |
| Mengesha et al., 2023 | Y | Y | Y | Y | Y | Y | Y | Y | Y | Y | Y | Y | Y | Y | Y | Y | 16 |
| Desta et al., 2021 | Y | Y | Y | Y | Y | Y | Y | Y | Y | Y | Y | Y | Y | Y | Y | N | 15 |
| Ayenew et al., 2020 | Y | Y | Y | Y | Y | Y | Y | Y | Y | Y | Y | Y | Y | Y | Y | Y | 16 |
| Alamneh et al., 2020 | Y | Y | Y | Y | Y | Y | Y | Y | Y | Y | Y | Y | Y | Y | Y | Y | 16 |
| Omolo et al., 2023 | N | Y | Y | Y | N | Y | Y | Y | Y | Y | Y | Y | Y | Y | Y | Y | 15 |
| Bogale et al., 2021 | Y | Y | Y | Y | Y | Y | Y | Y | Y | Y | Y | Y | Y | Y | Y | Y | 16 |
| Dessalegn Mekonnen, 2020 | Y | Y | Y | Y | Y | Y | Y | Y | Y | Y | Y | Y | Y | Y | Y | Y | 16 |
| Kassie et al., 2020 | Y | PY | Y | Y | Y | Y | Y | Y | Y | Y | Y | Y | Y | Y | Y | Y | 15 |
| Yirsaw et al., 2024 | Y | Y | Y | Y | Y | N | Y | Y | Y | Y | Y | Y | Y | Y | Y | Y | 16 |
| Lakew et al., 2024 | Y | Y | Y | Y | Y | N | Y | Y | Y | Y | Y | Y | Y | Y | Y | Y | 16 |

*Y=Yes, PY = Partial yes, N = No.

## Results

### Literature searches

A total of 722 studies were retrieved through all searching databases, including Google Scholar; 118 duplicate records were removed, and the remaining 604 studies were eligible for further screening. However, the majority of studies (n = 547) were excluded by screening their titles and abstracts. Then, the remaining 57 full-text articles were examined for eligibility criteria, and 46 studies were dropped for different reasons, such as variation in the study context, insufficient data, not directly related to the outcome of interest, scoping reviews, and qualitative systematic reviews. Finally, 11 eligible SRM studies were included for the final quantitative meta-analysis [21,22,25,26,30–36] (Fig 1).

### Characteristics of the included studies

In this umbrella review, 11 eligible SRM articles with a total of 143,327 participants were included. In terms of the distribution of the SRM studies across Africa, seven studies were conducted in Ethiopia, one in Botswana, and the remaining were conducted in the SSA context (Table 2).

### Uptake of cervical cancer screening in Africa

This umbrella review comprised 11 systematic reviews and meta-analyses from previous studies looking at the overall prevalence of cervical cancer screening in Africa. Thus, the pooled prevalence of cervical cancer screening uptake in Africa was 20.94% (95% confidence interval: 15.84%−26.04%). The statistical test found significant heterogeneity among the included SRM studies (heterogeneity $I^2$ = 99.88%, p-value = 0.000). As a result, a random-effects meta-analysis model was applied (Fig 2).

### Heterogeneity and sub-group analysis

Sub-group analysis was carried out based on the region of Africa, which the study was conducted and the study population. Accordingly, the highest prevalence was found in SSA, 31.44% (95% CI: 18.02, 44.87), whereas the lowest prevalence was detected in East Africa, 14.97% (95% CI: 10.39, 19.54) (Fig 3).

In addition, a subgroup analysis was carried out based on the study population; relatively, a higher overall uptake of cervical cancer screening was found among women with HIV (23.37%; 95% CI: 16.42, 30.31) Despite conducting

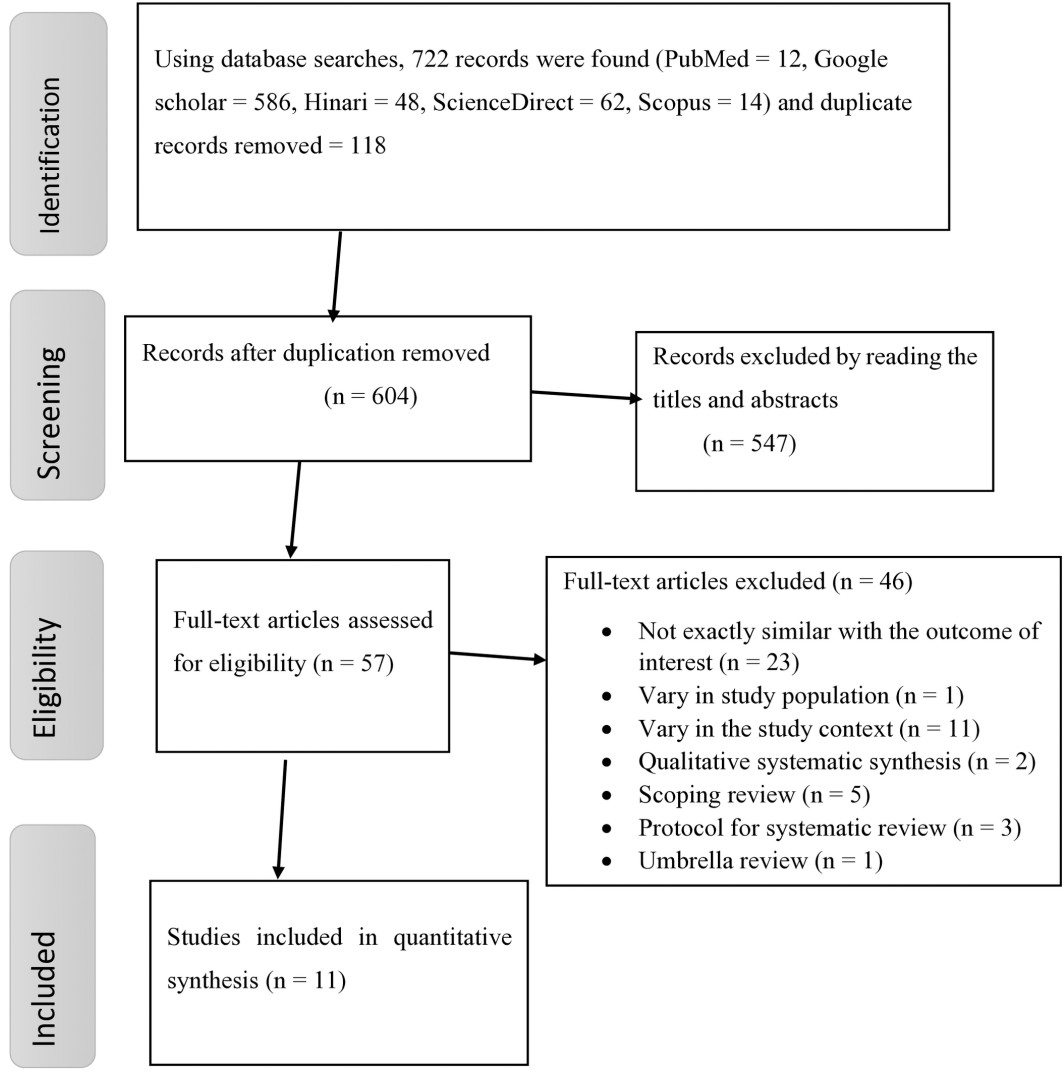

**Fig 1. PRISMA flow diagram showing the selection of studies for the umbrella review of cervical cancer screening uptake in Africa.**

subgroup analysis based on the aforementioned factors, no significant improvement in heterogeneity in the pooled estimate of cervical cancer screening uptake (Fig 4).

## Publication bias, trim and fill analysis

A funnel plot was used to visually check the presence of publication bias, while Egger's test was employed to confirm it. In this review, the asymmetrical distribution of the funnel plot demonstrates an existence of publication bias among the studies used to estimate the pooled prevalence of cervical cancer screening among African women. Statistically, Egger's (P-value = 0.000) and Begg's tests (P-value = 0.0430) were statistically significant, indicating the presence of publication bias (Fig 5).

A non-parametric trim and fill statistical analysis was performed to determine the number of potentially missing studies to reduce and adjust for publication bias in the included studies. However, the trim and fill analysis revealed the absence

**Table 2.** Descriptive summary of included systematic review and meta-analysis studies for uptake of cervical cancer screening and its determinants in Africa.

| Authors' name | Study area | Study population | Number of studies | Sample size | CCS uptake (%) | AMSTAR-2 score |
|---|---|---|---|---|---|---|
| Yimer et al., 2021 [30] | SSA | All women | 29 | 36,374 | 12.87 | 15 |
| Mengesha et al., 2023 [26] | SSA | HIV-positive women | 21 | 20,672 | 30 | 16 |
| Desta et al., 2021 [31] | Ethiopia | Eligible women | 25 | 18,067 | 14.79 | 15 |
| Ayenew et al., 2020 [32] | Ethiopia | Eligible women | 24 | 14,582 | 13.46 | 16 |
| Alamneh et al., 2020 [36] | Ethiopia | Reproductive age group | 18 | 9897 | 14.02 | 16 |
| Omolo et al., 2023 [33] | Botswana | Eligible women | 9 | 3,398 | 40 | 15 |
| Bogale et al., 2021 [22] | SSA | HIV positive women | 8 | 2,186 | 43 | 16 |
| Desalegn Mekonnen, 2020 [34] | Ethiopia | HIV positive women | 7 | 2822 | 18.17 | 16 |
| Kassie et al., 2020 [21] | Ethiopia | All women | 44 | 28,186 | 5.47 | 15 |
| Yirsaw et al., 2024 [25] | Ethiopia | HIV positive women | 12 | 4,905 | 21 | 16 |
| Lakew et al., 2024 [35] | Ethiopia | Female health professionals | 7 | 2,238 | 18 | 16 |

Key: CCS–Cervical cancer screening, SSA: Sub-Saharan Africa.

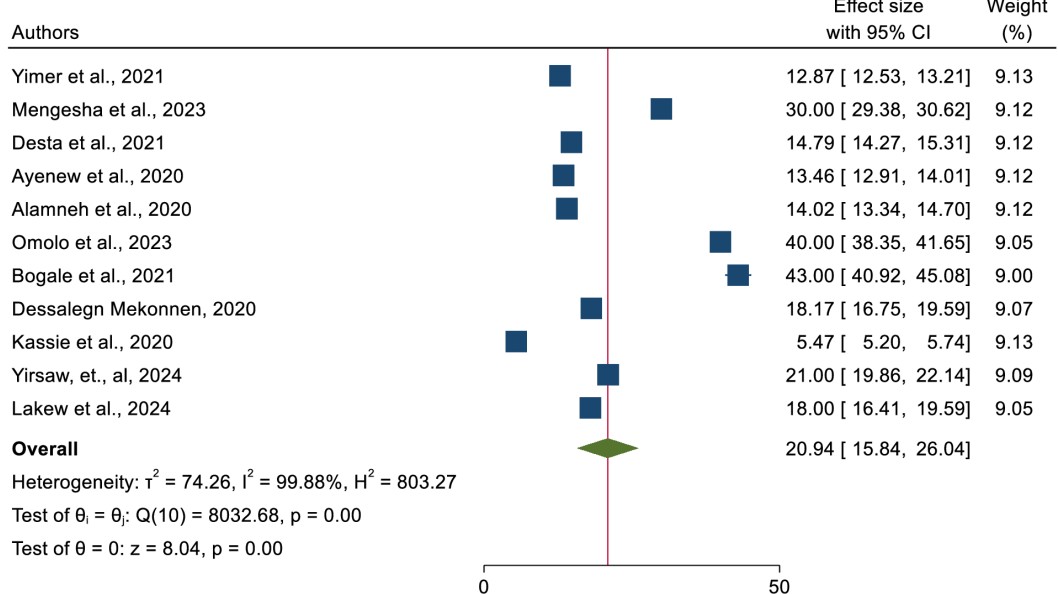

Random-effects DerSimonian–Laird model

**Fig 2. A forest plot for systematic reviews and meta-analyses of studies that showing the pooled prevalence of cervical cancer screening uptake in Africa.**

of significant publication bias because the overall prevalence of observed studies was equal to the sum of observed and imputed studies (Table 3).

## A leave-one-out sensitivity analysis

A leave-one-out sensitivity analysis using the random-effects model was performed to examine the effect of a single study on the estimated effect size. However, the findings show that a single study did not significantly affect the total effect size,

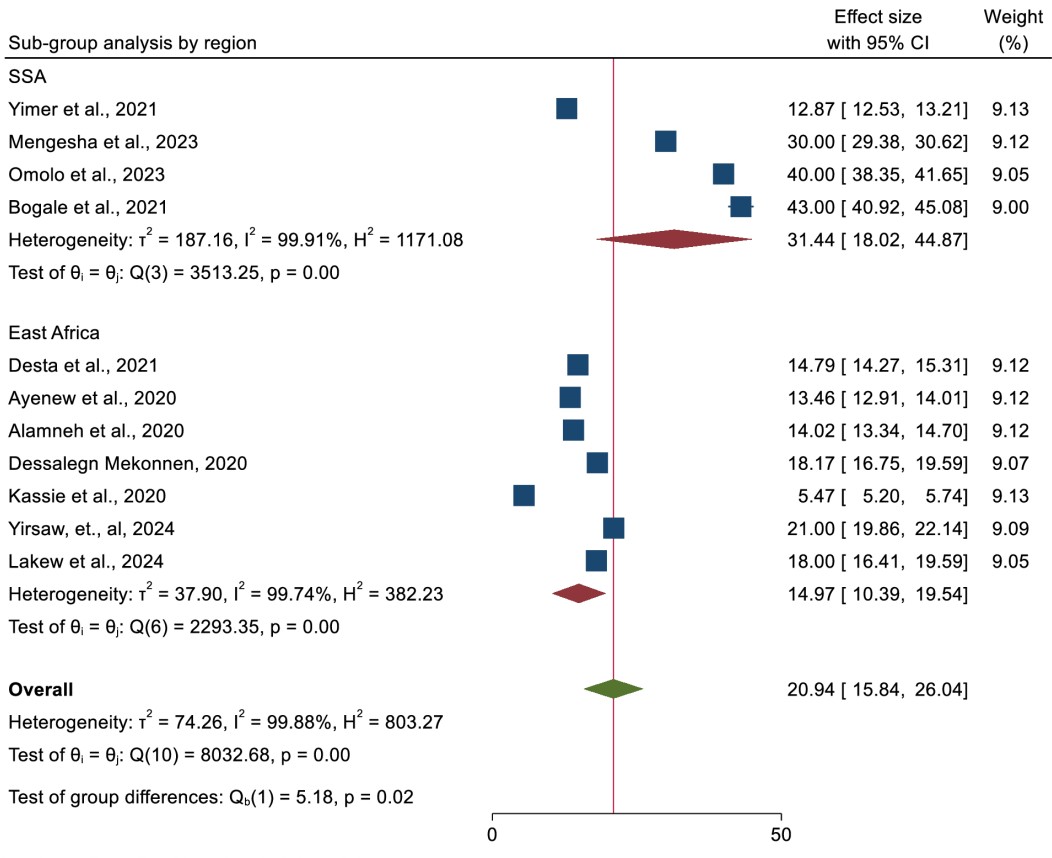

**Fig 3. A Forest plot showing the sub-group analysis of uptake of cervical cancer screening by region of Africa.**

and the point estimate of omitted study falls within the confidence interval of the overall estimate of CCS. This proved the reliability of the pooled estimate of cervical cancer screening in Africa (Fig 6).

## Factors associated with cervical cancer screening in Africa

This umbrella review examined eight publications out of eleven studies that reported on determinants that influence cervical cancer screening uptake in Africa. The random-effects model revealed that women's knowledge of cervical cancer screening, attitudes toward CCS, perceived vulnerability to cervical cancer, and history of sexually transmitted infections (STIs) were all significantly associated with cervical cancer screening in Africa.

Those who had knowledge about cervical cancer screening were 3.5 times more likely to utilize cervical cancer screening (AOR = 3.47, 95% CI: 3.01, 4.17) than those who had poor knowledge about it. The odds of cervical cancer screening were 3.56 times more likely in women with a favorable attitude towards CCS (AOR = 3.56, 95% CI: 3.04, 4.16) than in women who had an unfavorable attitude towards the screening program. The pooled adjusted odds ratio of the three studies also showed that women who perceived higher risk of cervical cancer (AOR = 3.57, 95% CI: 2.75, 4.63) were 3.57 times more likely to adopt CCS. Finally, having a history of STIs was the other contributing factor for cervical cancer screening in Africa. Thus, the likelihood of undergoing CCS was five times higher among women with a history of sexually transmitted infections (AOR = 4.89, 95% CI: 3.14, 7.62) than their counterparts (Fig 7).

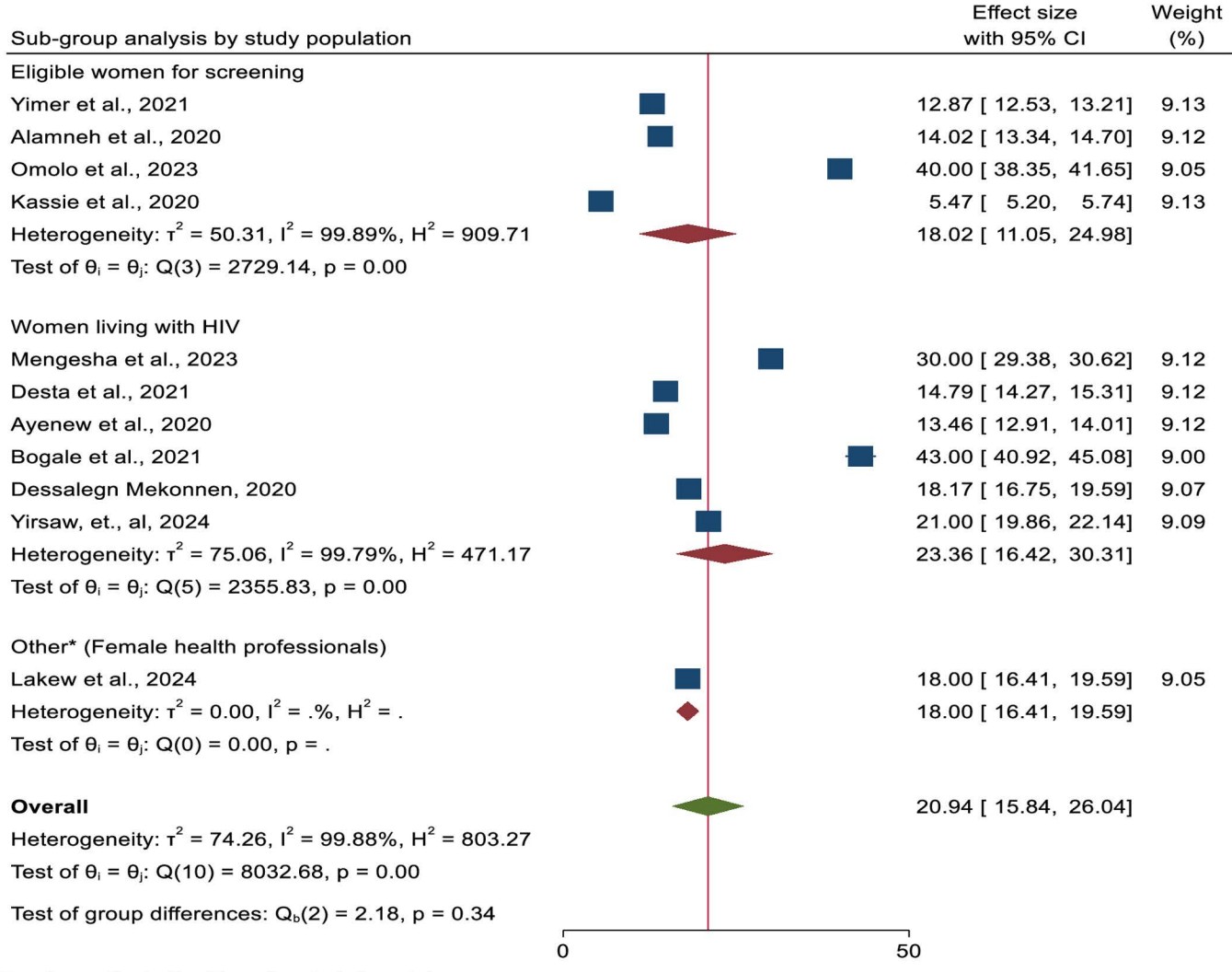

**Fig 4. A Forest plot showing the sub-group analysis of cervical cancer screening uptake in the context of Africa by study population.**

## Discussion

Secondary preventive measures, such as a high-coverage screening program, health promotion, and early detection of precancerous in treatment, are crucial for preventing cervical cancer, particularly in unvaccinated women and infected with HPV strains [16]. The prevalence of cervical cancer screening in resource-limited countries is significantly lower than that in developed countries [37]. Additionally, limited and inconsistent findings have been obtained regarding the use of cervical cancer screening services across Africa. Therefore, this umbrella review summarized prior meta-analysis studies on cervical cancer screening in Africa to generate comprehensive evidence.

In this umbrella review, the pooled prevalence of cervical cancer screening uptake in Africa was 20.94% (95% CI: 15.84%–26.04%). This finding was consistent with systematic review and meta-analysis studies conducted in 22% of LMICs [38], 18.2% in Arab countries [39], and 16% in Nepal [40]. However, the uptake of CCS in Africa was lower than systematic reviews and meta-analyses of studies that reported the prevalence of 46% among immigrant women in Europe [41] and 28% in Asia [42].

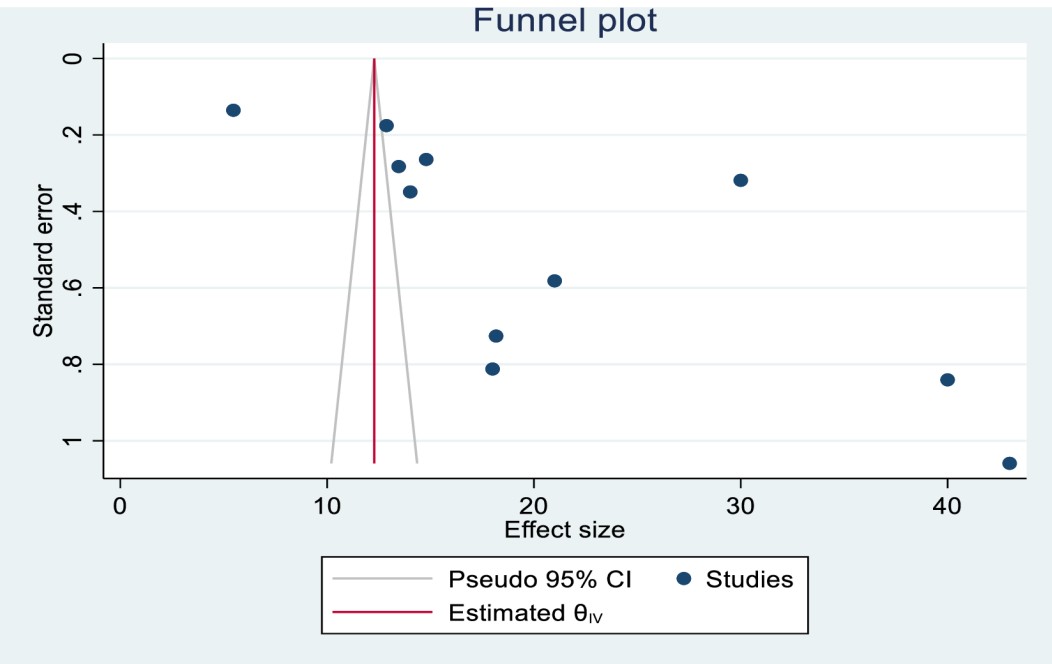

**Fig 5. A funnel plot test that demonstrating the prevalence of cervical cancer screening uptake in Africa.**

**Table 3. Non-parametric trim and fill analysis of publication bias for uptake of cervical cancer screening in Africa.**

| Studies | Number of studies | Effect size [95% confidence interval] | | |
|---|---|---|---|---|
| Observed | 11 | 20.940 | 15.836 | 26.044 |
| Observed+ Imputed | 11 | 20.940 | 15.836 | 26.044 |

Additionally, the finding of this umbrella review was lower than a global adherence to CCS that reported a pooled prevalence of 33.66%, with the lowest and highest adherence to CCS in LMICs and high-income countries, 24.91% and 75.66%, respectively [43]. The disparity could be explained by variations in the study participants' sociodemographic and financial situations. Thus, individuals who have a lower educational and socioeconomic background are less likely to adhere to preventative actions like cervical cancer screenings [43,44]. Moreover, variations in the countries' healthcare policies, such as the structure of institutions and the availability of screening programs, might have had an important effect on the implementation of effective cervical cancer screening initiatives [44]. Thus, low- and middle-income countries, particularly African countries, faced constrained opportunity for healthcare services, a shortage of skilled health workers, inadequate funds and supplies, and long waiting times. These are common barriers that lower uptake of CCS [44]. Likewise, lack of awareness, fear of the screening procedure, and unfavorable perceptions towards cervical cancer screening highly contribute for lower utilization of CCS in Africa. In contrast, individuals in the high-income countries had access to secondary preventive measures such as high-quality screening methods (Pap smear, HPV self-sampling, artificial-assisted screening), higher health literacy, and favorable attitudes towards CCS, which all ultimately increase the uptake of CCS [44,45]. Due to the low coverage of cervical screening, there is a higher rate of cervical cancer cases and poor prognosis related to late diagnosis in developing countries. Therefore, improving access to CCS, community-based health education, involving community leaders and health extension workers to overcome cultural and social hurdles, involving partners, making sure screening services are affordable, and integrating the program with other services should be the prioritized initiatives to increase CCS utilization.

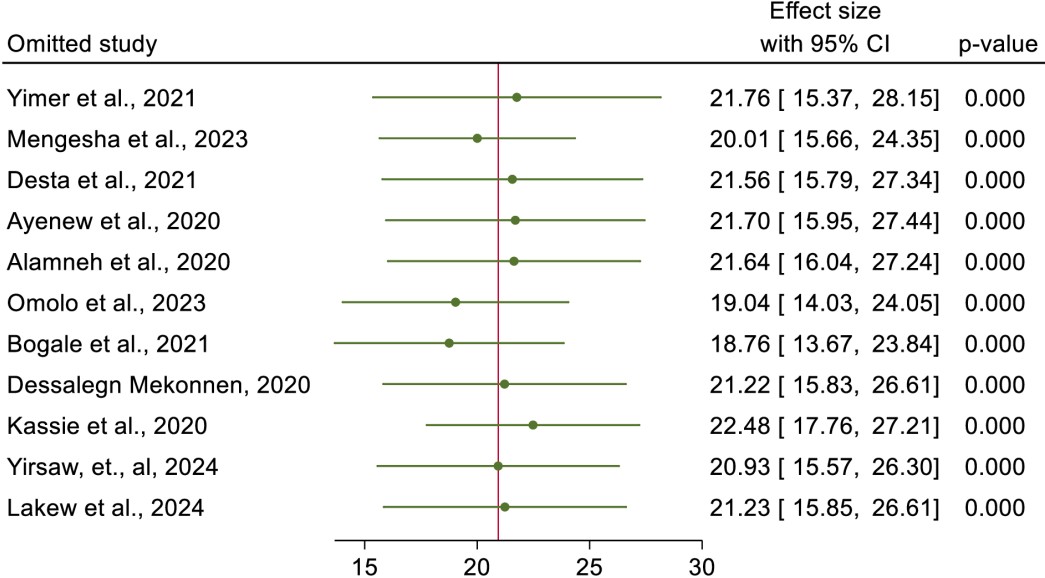

| Omitted study | | Effect size with 95% CI | p-value |
|---|---|---|---|
| Yimer et al., 2021 | | 21.76 [ 15.37, 28.15] | 0.000 |
| Mengesha et al., 2023 | | 20.01 [ 15.66, 24.35] | 0.000 |
| Desta et al., 2021 | | 21.56 [ 15.79, 27.34] | 0.000 |
| Ayenew et al., 2020 | | 21.70 [ 15.95, 27.44] | 0.000 |
| Alamneh et al., 2020 | | 21.64 [ 16.04, 27.24] | 0.000 |
| Omolo et al., 2023 | | 19.04 [ 14.03, 24.05] | 0.000 |
| Bogale et al., 2021 | | 18.76 [ 13.67, 23.84] | 0.000 |
| Dessalegn Mekonnen, 2020 | | 21.22 [ 15.83, 26.61] | 0.000 |
| Kassie et al., 2020 | | 22.48 [ 17.76, 27.21] | 0.000 |
| Yirsaw, et., al, 2024 | | 20.93 [ 15.57, 26.30] | 0.000 |
| Lakew et al., 2024 | | 21.23 [ 15.85, 26.61] | 0.000 |

Random-effects DerSimonian–Laird model

**Fig 6. A one-leave-out analysis for uptake of cervical cancer screening and its determinants in Africa.**

Regarding determinant factors, women's knowledge was found to be significantly associated with the acceptance of CCS screening; that is, women who knew more about cervical cancer and screening programs were more likely to use the CCS. This finding was consistent with previous studies [40,46,47]. This could be explained by their increased awareness of the importance, benefits, and screening process, which leads to increased participation and early detection of cervical cancer.

Cervical cancer screening was substantially correlated with women's attitudes toward CCS. As a result, women who had a favorable perception of CCS were more likely to get screened for cervical cancer than those who had a negative opinion. This result was in agreement with findings from low- and middle-income nations [11,48,49]. This could be because women with a good attitude have positive beliefs, motives, and views about the importance and benefits of regular screening, which in turn influence their screening practices. Cultural, social, and spiritual impediments, such as screening restrictions, fear of screening, shame, misunderstandings, and disapproval by partners or family members, all impede cervical cancer screening.

Women's perceived vulnerability for developing cervical cancer was the other contributing factor for cervical cancer screening. Thus, a woman who perceived herself as more vulnerable to cervical cancer had a higher likelihood of being screened for cervical cancer. This conclusion was in agreement with other studies [50,51]. This is because the women could recognize the severity and far-reaching consequences of the disease, which motivated them to get screened on a regular basis. However, it's crucial to increase the awareness, attitude, and practice of individuals as well as the community on the benefits of CCS and the severity of cervical cancer. It encourages undergoing screening, which enables detecting and treating the problem early and reducing the incidence of advanced-stage cervical cancer. Finally, women who had a history of STIs were five times more likely to undergo cervical cancer screening than those who did not have a history of STIs. This finding was consistent with primary studies conducted in Ethiopia [52–54] and a meta-analysis study conducted in Ethiopia [31,51]. This is due to the fact that women who have a history of sexually transmitted infections are more likely to visit healthcare facilities because they seek medical treatment and follow-up care for their illness. This allows health professionals to undergo cervical cancer screening for women as part of their comprehensive care for reproductive and sexual health.

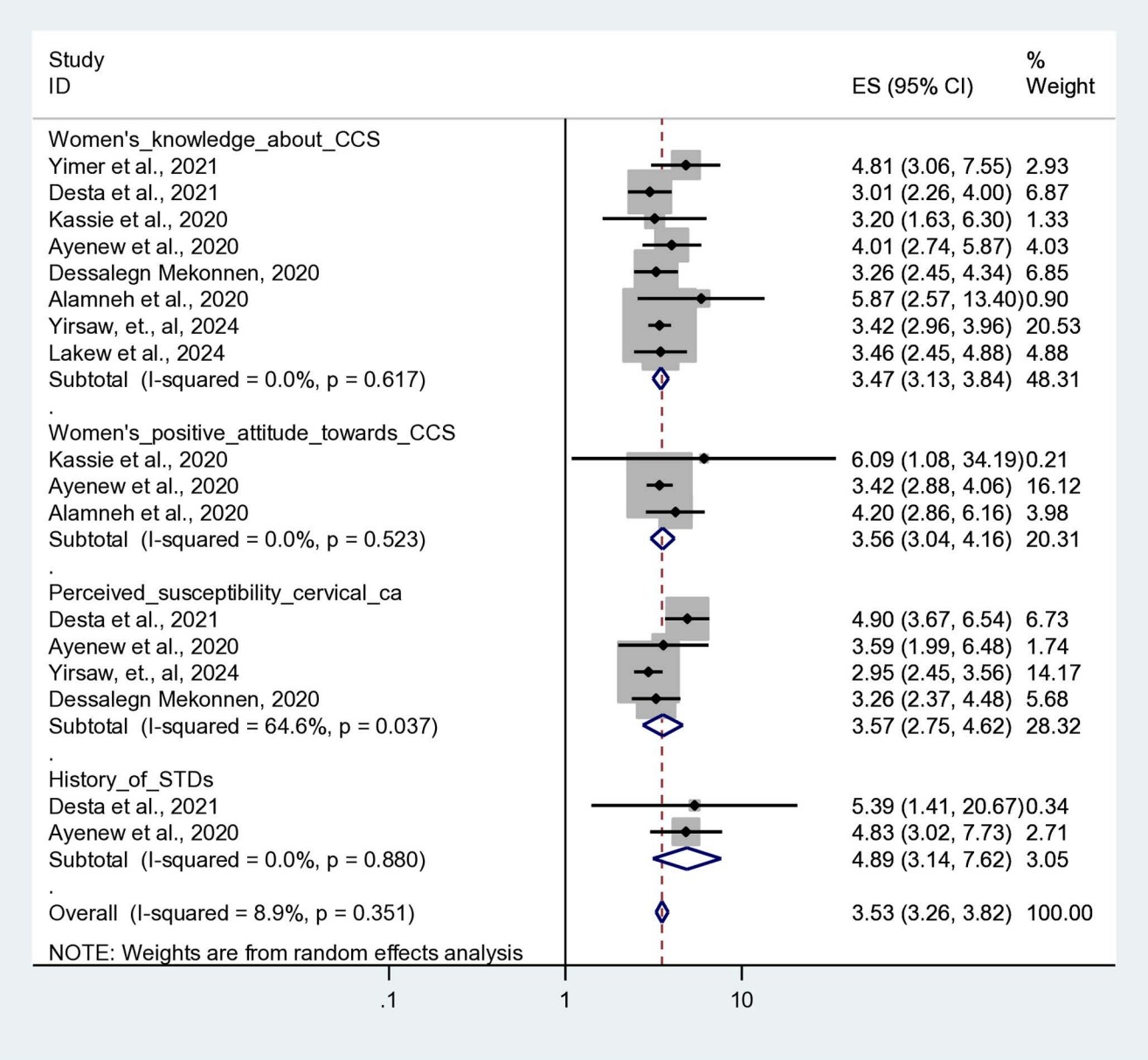

**Fig 7. A Forest plot that shows the pooled adjusted odds ratio (AOR) of factors associated with uptake of cervical cancer screening uptake in Africa.**

## Implication of the study

This umbrella review would be the pioneer to summarize preexisting multiple systematic reviews to generate comprehensive evidence on cervical cancer screening and its determinants in Africa. It highlights that the uptake of CCS is too far from the WHO target. Therefore, the findings of this review serve as baseline information for health professionals, health managers, researchers, and concerned non-governmental organizations to take evidence-based interventions to improve CCS in Africa. It also provides an insight for policymakers to realize this lower uptake of CCS, but the higher burden of

cervical cancer in Africa. The finding suggests the need to draft and implement appropriate strategies that help to promote cervical cancer screening uptake. The government of each country needs to expand and advocate cervical cancer screening programs through facility- and community-based screening campaigns and convey information via various mass media. The above actions play an important role in early detection and treatment of precancerous cervical lesions and reduce the incidence of cervical cancer.

### Strengths and limitations of the study

The authors of this umbrella review employed robust statistical approaches, including a random-effects model, to determine pooled prevalence and account for heterogeneity across studies. The model also examined for publication bias statistically and inspected funnel plots visually. However, this finding could be biased due to the significant variability of the included studies across Africa. This might be due to differences in the study area and the methodological quality of the original studies (sampling technique, sample size, and CCS uptake measurement) included in each systematic review and meta-analysis, which could alter the pooled estimate's generalizability. Additionally, it's better to explore barriers and facilitators of cervical cancer screening through qualitative systematic reviews or in-depth qualitative studies. This would be pivotal in exploring socio-cultural variables, health system variables, women's knowledge, and perceptions towards CCS.

## Conclusions

This umbrella review concludes that the pooled estimate of cervical cancer screening in Africa (20.94%) remains much lower than the WHO recommendations target (70%). In addition, women's level of knowledge, attitude toward CCS, perceived susceptibility to cervical cancer, and history of STIs were significantly associated with the uptake of cervical cancer screening. Therefore, to improve cervical cancer screening uptake, healthcare professionals, policymakers, and healthcare stakeholders shall implement effective strategies such as empowering women, improving knowledge and attitude towards cervical cancer screening, and offering CCS for a woman with a history of STIs.

## Supporting information

**S1 File. PRISMA 2020 Checklist.**
(DOCX)

**S2 File. Searching strategies.**
(PDF)

**S3 File. Excel data extraction.**
(XLSX)

## Acknowledgments

We would like to express our heartfelt gratitude to all of the authors of the primary and systematic reviews and meta-analyses included in this umbrella review.

## Author contributions

**Conceptualization:** Berihun Agegn Mengistie, Mihret Melese, Amlaku Nigusie Yirsaw.

**Data curation:** Berihun Agegn Mengistie, Mihret Melese, Amare Belete Getahun, Habtu Kifle Negash, Alemken Eyayu Abuhay.

**Formal analysis:** Berihun Agegn Mengistie, Ashebir Mamay Gebiru, Mihret Getnet, Amare Belete Getahun, Amlaku Nigusie Yirsaw.

**Methodology:** Berihun Agegn Mengistie, Mihret Getnet, Mikias Mered Tilahun, Desale B. Asmamaw, Desalegn Anmut Bitew.

**Software:** Ashebir Mamay Gebiru, Worku Chekol Tassew, Mikias Mered Tilahun, Desalegn Anmut Bitew.

**Visualization:** Yosef Belay Bizuneh, Habtu Kifle Negash, Alemken Eyayu Abuhay.

**Writing – original draft:** Worku Chekol Tassew, Nebebe Demis Baykemagn, Desale B. Asmamaw.

**Writing – review & editing:** Berihun Agegn Mengistie, Mihret Melese, Yosef Belay Bizuneh, Nebebe Demis Baykemagn.

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
