## [Decision Letter · Decision Letter 0]

Dear Dr. Mengistie,

Thank you for submitting your manuscript to PLOS ONE. After careful consideration, we feel that it has merit but does not fully meet PLOS ONE’s publication criteria as it currently stands. Therefore, we invite you to submit a revised version of the manuscript that addresses the points raised during the review process.

We look forward to receiving your revised manuscript.

Kind regards,

Worku Necho Asferie, MSc

Academic Editor

PLOS ONE

https://doi.org/10.1186/s13643-018-0874-7

https://doi.org/10.1016/j.puhe.2021.03.022

In your revision ensure you cite all your sources (including your own works), and quote or rephrase any duplicated text outside the methods section. Further consideration is dependent on these concerns being addressed.

4. As required by our policy on Data Availability, please ensure your manuscript or supplementary information includes the following:

5. We note that your Data Availability Statement is currently as follows: [All relevant data are within the manuscript and its Supporting Information files.]

Reviewers' comments:

Reviewer's Responses to Questions

**Comments to the Author**

1. Is the manuscript technically sound, and do the data support the conclusions?

Reviewer #1: Yes

2. Has the statistical analysis been performed appropriately and rigorously?

Reviewer #1: Yes

3. Have the authors made all data underlying the findings in their manuscript fully available?

Reviewer #1: Yes

4. Is the manuscript presented in an intelligible fashion and written in standard English?

Reviewer #1: No

Reviewer #1: Summary: This is a well-performed systemic review (umbrella review) of systemic reviews and meta-analyses on the prevalence of cervical cancer screening uptake in Africa and factors associated with screening uptake. The study was comprehensive and a thorough effort was put into understanding the accuracy and validity of the data to make claims for a large region. The methods and data could be presented more clearly and concisely. The manuscript would benefit from a discussion of the implications of these findings and suggestions or recommendations for future interventions or policies to improve cervical cancer screening implementation. Finally, the paper would benefit from a critical read for typographical errors, grammatical edits, and overall flow and clarity, which might limit its impact.

Major Comments

• Methods: Additional clarity about how the authors measured the outcome variable would be beneficial. How were the aOR from the prior reports included and combined? Are the aOR pooled estimates? Do they take into account other factors (ie- multivariable)? This is not clearly described in the methods.

• In the discussion, there are suggestions as to why there is lower screening prevalence in Africa and why specific factors are correlated with higher uptake. These suggestions should be based on supporting data either from the articles reviewed or other sources.

• The discussion section would be improved by a more comprehensive discussion about the implications of the identified factors which increased uptake. Are there recommendations or suggested next steps to help move the field forward? For example, instead of expanding on why higher perceived risk of cervical cancer increases rates of screening, discuss how to improve individuals’ understanding of the risks of cervical cancer as a tool to increase screening or other suggestions with these findings. This should be a significant portion of the discussion and conclusions.

• Recommend discussion of limitations of this paper, namely the focus on uptake by individuals rather than providers or systems. This is an important component of the implementation of screening, but it should be mentioned that there are other barriers beyond getting individual patients to agree to screening.

• The paper would benefit from a critical read for typographical errors, grammatical edits, and overall flow and clarity, which might limit its impact (many suggestions are outlined below).

Minor Comments

• While abbreviations are defined at the end, it would be significantly easier to read if they are defined at first use.

o For example, CCS and SRM should be defined on first use in abstract and Introduction/Methods, since these are non-standard abbreviations and key to understanding manuscript.

• Page 4 Introduction

o Avoid using contraction, should be “Although it is a global health problem…”

o HPV sub type info is not relevant to remainder of paper

o Is there a citation for cervical HPV being most prevalent sexually transmitted infection?

• Page 5:

o Oxford comma would clarify 3rd paragraph: “The screening methods include cytological-based testing (pap smear slide cytology and liquid based-cytology, HPV DNA testing, and visual inspection tests”

o Would be helpful to note that while pap smear requires a Pathologist, other screening methods are available at the point of care and are appropriate for use in low-resource settings.

• Page 6:

o The methods state this was a review of papers on HPV vaccine uptake, but it is for cervical cancer screening.

• Page 7:

o Would re-order the sentences of the paragraph which starts on page 6 and ends on page 7 to list steps in the order they were performed. This will make it more clear to read (ie search terms should come before snowball sampling)

o Inclusion and exclusion criteria and study selection process can be included in the search strategy section without needing their own sections

o Would move measurement of outcome variable section to after data extraction

• Page 8

o Cite software the first time it is mentioned

o There is repetition in the Data extraction section from the Search strategy section. Only need to describe this process once.

o There are multiple sentences written in future tense which need to be changed to past tense

o I do not understand the sentence “the specific included article was extracted based on the 16 times of the AMSTAR 2 tool.” Can this be phrased differently for more clarity?

• Page 10

o Ethiopia and Botswana are part of Sub-Saharan Africa, so please clarify what what the difference is for the location of study sites. Were the studies “in SSA context” performed across multiple countries? If so, say this.

o In the sub-group analysis, it would be helpful to clarify the difference between East Africa and Sub-Saharan Africa since these are overlapping regions.

• Page 11

o Recommend combining sections “Publication bias,” “trim and fill analysis,” and “leave out sensitivity analysis” into one section, especially since there is disagreement between the methods for publication bias.

o Avoid using the abbreviation STD which is not used in remainder of text, and stick with STI.

• Page 12

o Would remove discussion of this data compared to Nepal and the Philippines unless specifically comparing to other regions – they are both mentioned in the context of other data on African countries.

**Do you want your identity to be public for this peer review?** For information about this choice, including consent withdrawal, please see our Privacy Policy

Reviewer #1: No

---

## [Author Response · Author response to Decision Letter 1]

3 Feb 2025

The authors would like to thank the PLOS ONE Academic Editorial Team, the Editor who handle this manuscript, and the reviewers who have been reviewing, giving swift and valuable comments and suggestions to improve the quality of the manuscript. We have thoroughly considered every suggestion provided, and the corresponding revisions have been diligently incorporated into the updated version of the manuscript. Your expertise and guidance have been pivotal in enhancing the overall quality of our manuscript. We have incorporated all minor and major comments and suggestions of Academic Editor and Reviewers with track changes in the revised manuscript and a point-by-point response. We kindly invite to take a look file attached as "Response to Reviewers"

---

## [Editor Report · Decision Letter 1]

Dear Dr. Mengistie,

Thank you for submitting your manuscript to PLOS ONE. After careful consideration, we feel that it has merit but does not fully meet PLOS ONE’s publication criteria as it currently stands. Therefore, we invite you to submit a revised version of the manuscript that addresses the points raised during the review process.

We look forward to receiving your revised manuscript.

Kind regards,

Worku Necho Asferie, MSc

Academic Editor

PLOS ONE
---

## [Author Response · Author response to Decision Letter 2]

1 Apr 2025

Dear Academic Editor, we have addressed the comments and all the changes are highlighted in the Tracked change and included in the revised manuscript. In addition to this, we kindly request you to check the point-by-point response file for the raised comments.

Thank you in advance!

---

## [Editor Report · Decision Letter 2]

Uptake of cervical cancer screening and its determinants in Africa: Umbrella review

PONE-D-24-50654R2

Dear Dr. Mengistie,

We’re pleased to inform you that your manuscript has been judged scientifically suitable for publication and will be formally accepted for publication once it meets all outstanding technical requirements.

Kind regards,

Worku Necho Asferie, MSc

Academic Editor

PLOS ONE
---

## [Editor Report · Acceptance letter]

PONE-D-24-50654R2

PLOS ONE

Dear Dr. Mengistie,

I'm pleased to inform you that your manuscript has been deemed suitable for publication in PLOS ONE. Congratulations! Your manuscript is now being handed over to our production team.

Kind regards,

on behalf of

Assistant Professor Worku Necho Asferie

Academic Editor

PLOS ONE